# Identifying Defects without *a priori* Knowledge in a Room-Temperature Semiconductor Detector Using Physics Inspired Machine Learning Model

**DOI:** 10.3390/s24010092

**Published:** 2023-12-23

**Authors:** Srutarshi Banerjee, Miesher Rodrigues, Manuel Ballester, Alexander Hans Vija, Aggelos Katsaggelos

**Affiliations:** 1Department of Electrical and Computer Engineering, Northwestern University, Evanston, IL 60208, USA; manuelballestermatito2021@u.northwestern.edu (M.B.); a-katsaggelos@northwestern.edu (A.K.); 2Siemens Medical Solutions USA, Inc., Hoffmann Estates, IL 60192, USA; miesher@umich.edu (M.R.); hans.vija@siemens-healthineers.com (A.H.V.)

**Keywords:** room temperature semiconductor detector, physics inspired machine learning model (PI-ML), machine learning, material characterization, charge transport, trapping, detrapping, trapping centers, defects

## Abstract

Room-temperature semiconductor radiation detectors (RTSD) such as CdZnTe are popular in Computed Tomography (CT) imaging and other applications. Transport properties and material defects with respect to electron and hole transport often need to be characterized, which is a labor intensive process. However, these defects often vary from one RTSD to another and are not known *a priori* during characterization of the material. In recent years, physics-inspired machine learning (PI-ML) models have been developed for the RTSDs which have the ability to characterize the defects in a RTSD by discretizing it volumetrically. These learning models capture the heterogeneity of the defects in the RTSD—which arises due to the fabrication process and the energy bands of elements in the RTSD. In those models, the different defects of RTSD—trapping, detrapping and recombination for electrons and holes—are present. However, these defects are often unknown. In this work, we show the capabilities of a PI-ML model which has been developed considering all the material defects to identify certain defects which are present (or absent). Additionally, these models can identify the defects over the volume of the RTSD in a discretized manner.

## 1. Introduction

With the growth of nuclear technology, especially for radiation monitoring and sensing, radiation detectors are becoming vital for cutting-edge technologies. RTSDs are the leading radiation detectors which are widely used in research and industry alike. RTSDs have wide range of applications for high energy photons, starting from medical computed tomography systems to homeland security and others [1,2,3]. These applications call for image sensors with high quality crystals with uniform and optimized charge transport properties at a reasonable cost. Compared to Si and HPGe detectors, RTSDs have widespread use due to their higher stopping power and operation at high voltages due to their high resistivity. Additionally, RTSDs with uniform charge transport properties including no polarization, high breakdown voltage, high charge drift velocities, high energy resolution and excellent fabrication quality are desired. Several electrode configurations are possible for these radiation detectors depending on the end applications—cylindrical geometry, planar geometry and others. Often, they are used in pixelated anode patterns in compact radiation detector units.

The repeatable properties of RTSDs in a detector and across multiple detectors depends on the manufacturing process and other random factors. In large-area CdZnTe and CdTe detectors, spatial variations in response exists due to the crystalline defects such as Te inclusions, and local variations in the electric field [4]. RTSD characterization in terms of material and electrical properties are critical to achieve sub-millimeter position detection accuracy and energy resolution below 1.0% at 662 keV. Detailed characterization of a detector module is time consuming and also requires sophisticated experimental setups and skilled manpower. However, characterization of RTSD arrays for individual detectors and identifying the defects within the sensors spatially and temporally aids in developing novel imaging algorithms and allows re-use of these RTSD with inferior charge transport properties.

Under high photon fluxes ($10^{6}\;{\mathrm{mm}}^{-2}s^{-1}$), the RTSD (for example CdZnTe) is prone to polarization effects, where the internal electrical field is distorted due to build up of trapped charges [5]. Pixelated CdZnTe detectors are often used in pulse mode at high-fluxes [5,6] to avoid this effect. Recently developed CdZnTe [7] has been characterized using pulse shape analysis of the measured signals in high-flux scenarios [8] and the charge transport properties are estimated. Thermoelectric emission spectroscopy (TEES) and thermally stimulated conductivity (TSC) measurements were used in [9] to measure the thermal ionization energies of the electron and hole charge trap levels. TSC measurements for characterization of deep trap levels in CdZnTe has been done in [10] by simultaneous multiple peak analysis. The average trapping and detrapping times for holes were derived [11] using the average hole trapping time $\tau_{h}$ as measured in [12,13,14,15,16] using a statistical charge collection efficiency model based on known electron average trapping time. By direct comparison between measured and simulated signals at the cathode, average hole detrapping time $\tau_{dh}$ is obtained [17,18,19,20]. The influence of deposition methods and type of metal contacts on trapping/recombination defects at the metal/semiconductor interfaces is studied in [21]. The uniformity of high-flux CdZnTe is characterized in [4]. These classical approaches of measuring electron and hole trap properties provide an average behavior over the material and require cumbersome multiple experiments with high technical skills.

Machine learning and deep learning have revolutionized various science and engineering disciplines in the last decade or so. Applying machine learning to physical systems, material science, drug discovery and others has been the focus of recent research. Integrating physical models with machine learning has been popular over the last few years [22,23]. Solving partial differential equation (PDE)-based physical systems using Neural Networks has been done in [24,25]. A data-driven supervised technique to learn nonlinear operators is developed in DeepONets [26]. A Recurrent Neural Network has been used to model 2D wave equation [27], while Physics-Informed Neural Networks (PINNs) have been used to solve 2D Poisson equation [28]. Mostly, in these approaches, relatively simple PDEs have been solved. However, in RTSD, the charge transport involves a multitude of coupled PDEs [29] such as charge drift, trapping, detrapping and recombination of electrons and holes.

Applying machine learning to the modeling of RTSDs has been done in the last few years. In our previous works [30,31,32,33,34] we developed a PI-ML model for characterizing the RTSD. In this model, the physics-based charge transport equations are combined with a machine learning model to identify the material properties of the RTSDs. The classical physics-based equations for electron and hole charge transport are discretized over the volume of the RTSD. In each discretized volume (voxel), the different defects such as trapping, detrapping, recombination as well as electron and hole charge transport are formulated as unknown weights which are trained by the input electron–hole charge pair incident on the RTSD and the output signals and/or charges at the electrode on either ends of the RTSD. We developed the machine learning model considering the number of trainable weights as dictated by the physical laws [30], as well as a reduced set of trainable weights than what is dictated by the physical laws [31].

In these models, the defects of RTSD—trapping, detrapping and recombination for electrons and holes—are considered to be present. However, in practice for a RTSD, the presence or absence of these defects are often unknown. It not only depends on the type of RTSD, but also the fabrication process involved. In this work, we show the capabilities of a PI-ML model developed considering all the defects—trapping, detrapping and recombination—to identify certain defects which are present (or absent). Additionally, as in our earlier PI-ML models for characterizing RTSD, one can identify these defects over the volume of the RTSD in a discretized manner.

## 2. Materials and Methods

In RTSDs, electron and hole transport properties play an important role in selecting detectors for any application. Shallow and deep defects due to trapping, detrapping and recombination are governed by the Shockley–Read–Hall (SRH) Theory [35,36]. The macroscopic equations in [29] describe the various phenomena occurring in the material to produce electron–hole pairs when high-energy photons (X-rays or γ-rays) interact with the material usually by Photoelectric, Compton or Pair-Production interactions. Subsequently in the RTSD, the free charges drift towards the respective electrodes, along with trapping, detrapping in the defect levels and recombination of free charges in the bulk of the material [37]. The temporal dynamics of free electrons and holes following the SRH model is affected by the trapping energy states in the bandgap [38]. The signals are collected at the electrodes on either end of the RTSD. We consider a single cathode and nine grid anodes on either end of the RTSD as in [30,31].

The training data for the PI-ML model have been developed using the classical charge transport equations [29,30] in MATLAB, using MATLAB 9.6, for the RTSD by fixing the material properties—electron and hole mobilities—with trapping, detrapping, recombination lifetimes and electric field in the RTSD [30]. This charge transport model has been developed for a 1D RTSD which is discretized virtually into small microscopic sub-volumes, termed as voxels. In each of the voxels, the physics of electron and hole charge transport—drift, trapping, detrapping and recombination—holds true for the charge input at any voxel location. Over the time steps, the signals are collected at the electrodes—cathode and pixelated anodes. Moreover, the free and trapped charges in the voxels are computed over time steps. Each time step and total number of time steps is defined *a priori*. The input data for training this PI-ML model consist of a pair of electron–hole charge incidents at known voxels. The signals with free and trapped charges over time are the output of the learning-based model [30].

In [30,31], all the defects of the RTSD have been considered to be present. Multiple trapping levels were considered for holes and only one trapping level was considered for electrons along with recombination for electrons and holes. We consider CdZnTe with non-uniform spatial properties with two trapping centers for holes and one trapping center for electrons along with recombination. The electron and hole mobility lifetime products of $0.01\;{\mathrm{cm}}^{2}/V$ and $10^{-4}\;{\mathrm{cm}}^{2}/V$ are used, respectively. Mean electron trapping and detrapping lifetimes of 2.5μs and 0.26μs are used in the model. For hole-trapping center 1, trapping and detrapping lifetimes of 0.260μs and 0.045μs are considered, respectively, while for hole-trapping center 2, the trapping and detrapping lifetimes of 0.127μs and 0.096μs are considered, respectively. The electron and hole recombination lifetimes of 100μs and 1μs are considered. For each of these coefficients (referred to as τ in general), we compute the number of charged particles (electrons and holes) remaining in that particular energy level, using the formula $N_{\mathrm{left}}=N_{0}e^{-t/\tau }$. Here, $N_{0}$ and $N_{\mathrm{left}}$ refer to the charges in the particular energy level at time t=0 and at time *t*, respectively. So, $N_{\mathrm{left}}/N_{0}$ refers to the fraction of charges in that particular energy level at time *t*. However, to demonstrate the capabilities of a general PI-ML models of RTSDs in identifying only defects which are actually present, we allow only certain defects to be present in our classical model and remove other defects. Specifically, we create the following cases in the classical model: (i) only trapping centers and no recombination, (ii) no trapping centers and only recombination and (iii) no trapping centers and no recombination. These are used during simulation of the training data for the machine learning model.

The PI-ML model has been developed based on the classical charge transport equations in a RTSD [30,31] in Python, using version Python 3.8.2. The RTSD is discretized into *N* voxels. In each of the *N* voxels, the material properties such as charge transport, trapping, detrapping and recombination are modeled as trainable weights. Instead of using a traditional Convolutional Neural Network or Fully Connected Network to model the charge transport, we develop a novel recurrent neural network as shown in Figure 1 and described in [30,31]. Tensorflow library, Tensorflow 2.3.0, in Python has been used in developing the PI-ML model. The PI-ML model has a minimum number of trainable weights compared to its counterparts in the machine learning literature. Each recurrent unit is similar to a gated recurrent unit (GRU) or long short-term memory (LSTM). The model is trained with simulated data generated using the charge transport equations in MATLAB. Backpropagation through time (BPTT) [39,40] is used for computing the gradients of the loss function with respect to the trainable weights in the model. The update of the weights is based on stochastic gradient descent method—ADAM optimization [41]. A learning rate of $5\times 10^{-4}$ with two momentum terms $\beta_{1}=0.9$ and $\beta_{2}=0.999$ is used for training the PI-ML model. The model is trained with data generated by considering certain defects: (i) only trapping centers and no recombination, (ii) no trapping centers and only recombination and (iii) no trapping centers and no recombination. The PI-ML model, however, has the complete physical phenomena of a charge transport—drift, trapping, detrapping and recombination of charges.

The PI-ML model in general can be designed considering either all the charge transport trainable weights, or a reduced set of weights. In [30], we developed the model considering all the charge transport parameters. In [31], the model was developed considering only a reduced set of trainable weights than what is dictated by physical laws, and trained with free electron and hole charges in the RTSD. However, in this work, we demonstrate our approach based on the Physical Model-3 of [31] and show that our model learns only the material defects which are present in the data from the classical model. As in Physical Model-3, the trapping coefficient for multiple trapping centers (hole in this paper) is characterized in the equivalent manner. In this work, the accuracy of the trained weights with respect to the ground truth weights (used in the classical model) is measured in terms of Root of Normalized Mean Squared Error (RNMSE) as described in [31].

Thus, the dependency of the PI-ML model of RTSD on the precise properties of the material is eliminated. One can develop these PI-ML models just based on the general charge transport equations. The model possesses the capability to identify the defects present in the material from these generalized models and the training data. In Section 3, we show the results of our PI-ML model trained on data containing certain defects only.

## 3. Results

In our earlier paper [31], we experimented with different physical models, gradually removing the amount of data used for training the models. The material properties such as drift, trapping, detrapping and recombination for both electrons and holes were kept fixed in the ground truth data. However, in the real world, the detector will have defects which can be attributed to recombination, trapping with different trapping centers, non-uniform electric field or a combination of them. In this section, we use different material properties and create defects of various types and use a physical model to identify these properties. Physical Model-3 of the paper [31] is used in this simulation study for illustration purposes. The material properties of the Physical Model-3 contain drift, trapping, detrapping and recombination of charges (electrons and holes). We compute the Root of Normalized Mean Squared Error (RNMSE) for the different trained material properties and also the mean RNMSE values of these properties as well.

### 3.1. No Defects and No Recombination

The ground truth data has been generated using the classical model with no trapping centers for electrons and holes, along with no recombination of electrons and holes in the bulk of the crystal. The Physical Model-3 is trained with this ground truth data and the trained weights are analyzed. Figure 2 shows the variation of electron transport coefficients over the voxels. The RNMSE error for the elctron transport coefficients is 0.0014. The e-h injections are for every 5 voxels from Voxel 9 until Voxel 59.

The electron and hole trapping coefficients are shown in Figure 3a,b, respectively. The model is correctly able to identify the trapping coefficients of 0 for both electrons and holes. For holes, both the trapping probabilities are learned as 0 from Voxels 1 to 59, while for electrons, only one trapping probability is learned as 0 from Voxels 9 upto 99, which is the correct ground truth parameter of the detector as well. The RNMSE values for the electron and hole trapping coefficients are $9.6517\times 10^{-4}$ and $1.6113\times 10^{-8}$, respectively. We show only the trapping coefficients for electrons and holes and not the detrapping coefficients, since, if the trapping coefficients for electrons and holes are 0, then the detrapping coefficients does not matter.

The electron and hole recombination coefficients are shown in Figure 4a,b, respectively. The model is correctly able to identify the recombination coefficients of 0 for both electrons and holes. For holes, the recombination probabilities are learned as 0 from Voxels 1 to 59, while for electrons, the recombination probabilities are learned as 0 from Voxels 9 up to 99, which are the correct ground truth parameters of the detector. The RNMSE values for the electron and hole recombination coefficients are $9.6538\times 10^{-4}$ and $1.2609\times 10^{-7}$, respectively. The mean RNMSE value for the trained coefficients (transport of electrons, trapping of electrons and holes along with recombination of electrons and holes) is $6.6\times 10^{-4}$, which is a small value.

### 3.2. Defects and No Recombination

The ground truth data have been generated using the classical model with one trapping center for electrons and two trapping centers for holes with no recombination of electrons and holes in the bulk of the crystal. The Physical Model-3 is trained with this ground truth data and the trained weights are analyzed. Figure 5 shows the variation of electron transport coefficients over the voxels. The RNMSE value for the electron transport coefficients is 0.0058. The e-h injections are for every five voxels from Voxel 9 until Voxel 59.

Figure 6a,b shows the electron and hole trapping coefficients, respectively. It is seen that the model can learn the piece-wise linearly uniform electron trapping coefficients from Voxel 9 to Voxel 99 as shown in Figure 6a. Similarly, for holes, the model can learn the equivalent hole trapping coefficients, which are piece-wise linearly uniform as well. The hole coefficients are learned from Voxel 59 all the way towards the cathode (left end) up to Voxel 1 as shown in Figure 6b. The RNMSE values for the electron and hole trapping coefficients are 0.5845 and 0.0510, respectively.

Similarly, Figure 7a,b shows the electron and hole detrapping coefficients, respectively. It is seen that the model can learn the piece-wise linearly uniform electron detrapping coefficients from Voxel 9 to Voxel 99 as shown in Figure 7a for most of the voxels. Around Voxel 40, there is a single position where the electron detrapping coefficient is learned incorrectly, but for other voxels the coefficients are learned correctly. Similarly, for holes, the model can learn the equivalent hole detrapping coefficients which are piece-wise linearly uniform as well. The hole coefficients are learned from Voxel 59 all the way towards the cathode (left end) up to Voxel 1 as shown in Figure 7b. The RNMSE values for the electron and hole detrapping coefficients are 1.0498 and 0.0728, respectively. We observe that for multiple hole trapping centres, the hole trapping and detrapping coefficients are learned as the equivalent trapping and detrapping coefficients—which are the characteristics of Physical Model-3 [31].

The electron and hole recombination coefficients are shown in Figure 8a,b respectively. The model is correctly able to identify the recombination coefficients of 0 for both electrons and holes. For holes, the recombination probabilities are learned as 0 from Voxels 1 to 59, while for electrons, the recombination probabilities are learned as 0 from Voxels 9 up to 99, which are the correct ground truth parameters of the detector. The RNMSE values for the electron and hole recombination coefficients are 0.0026 and $8.6241\times 10^{-5}$, respectively. The mean RNMSE value for the trained coefficients (transport of electrons, trapping and detrapping coefficients of electrons and holes along with recombination of electrons and holes) is 0.2524. This RNMSE value is attributed to the fact that one of the trained electron trapping and detrapping coefficients has a significantly high value near Voxel 40 and Voxel 100, which makes the RNMSE value high.

### 3.3. No Defects and with Recombination

The ground truth data have been generated using the classical model with no electron and hole trapping centers but with just recombination for both electrons and holes. The Physical Model-3 is trained with this ground truth data. Figure 9 shows the variation of electron transport coefficients over the voxels, with the e-h injections for every five voxels from Voxel 9 until Voxel 59. The RNMSE value for electron transport coefficients is 0.0025.

Figure 10a,b shows the electron and hole trapping coefficients, respectively, for the voxels trained by the multiple e-h injections. Both these trapping coefficients converge to 0, which is the ground truth parameter in the data. The RNMSE values for the electron and hole trapping coefficients are 0.0018 and 0.0011, respectively.

Similarly, Figure 11a,b shows the electron and hole recombination coefficients, respectively, for the voxels trained by the multiple e-h injections. The hole recombination coefficients converge to the ground truth parameters better than the electron recombination coefficients. For the electron recombination coefficients, the learned coefficients converge better for Voxels 9 to 34, but the rest of the coefficients converge to a value which is close to the ground truth recombination coefficients. The RNMSE values of the electron and hole recombination coefficients are 1.6937 and 0.1383, respectively. The mean RNMSE value for the trained coefficients (transport of electrons, trapping of electrons and holes along with recombination of electrons and holes) is 0.3675. The mean RNMSE value is significantly contributed due to the high value of the learned electron recombination coefficients near the anode (Voxel 100).

## 4. Discussion

We performed simulation experiments with different material properties as shown by removing properties such as trapping centers and recombination from the training data. We performed our experiments using Physical Model-3 as described in [31]. It is seen that despite the material properties which are absent not being explicitly specified in the physical model, our learning-model can identify the defects which are present or absent and can identify the magnitude of the defects as well. Thus, the learning models can identify the structure of the model from the data presented. This is highly beneficial, especially in cases where the material defects can be present or absent and it can vary spatially over the crystal and with time as well. A few (<5) trained electron coefficients deviate from the ground truth significantly, which makes the RNMSE value higher than expected. However, for most of the regions in the crystal, the trained coefficients converge perfectly to the ground truth parameters. We demonstrate this approach for Physical Model-3, which is trained just on free electron and hole charges in the RTSD. However, this is also valid for other PI-ML models in [31] as well, which consist of all the trainable weights as dictated by the physical laws or a reduced set of trainable weights.

## 5. Conclusions

The physical model is trained with the presence and absence of defects such as trapping centers and recombination. The model can correctly identify the presence or absence of these defects as well as find out the magnitude of such defects in the crystal spatially over the voxels. Spatially piece-wise uniform material properties are also used in this simulation. The Root of Normalized Mean Squared Error (RNMSE) values for these models has been computed and analyzed. The learning models show promising results which could lead the way for future developments in characterization of the RTSD with fewer experimental setups and data.

## Figures and Tables

**Figure 1 sensors-24-00092-f001:**
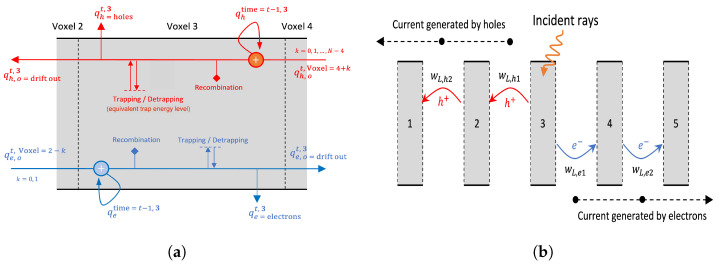
(**a**) PI-ML unit for one voxel. (**b**) PI-ML model with multiple voxels. While the charges drift towards the electrodes, signals are generated at the electrodes. Figure adapted with permission from [30,31].

**Figure 2 sensors-24-00092-f002:**
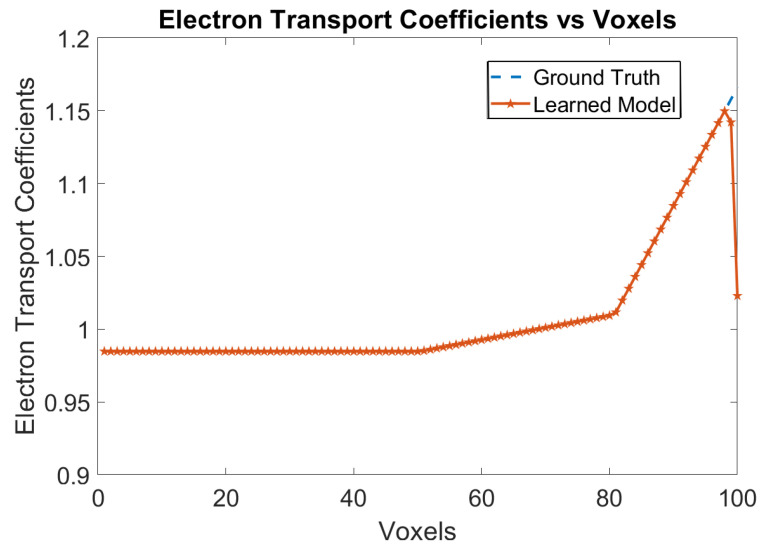
Electron transport coefficients using Physical Model-3 for e-h injections at Voxels 9, 14, 19, 24, 29, 34, 39, 44, 49, 54 and 59 for the model in Section 3.1.

**Figure 3 sensors-24-00092-f003:**
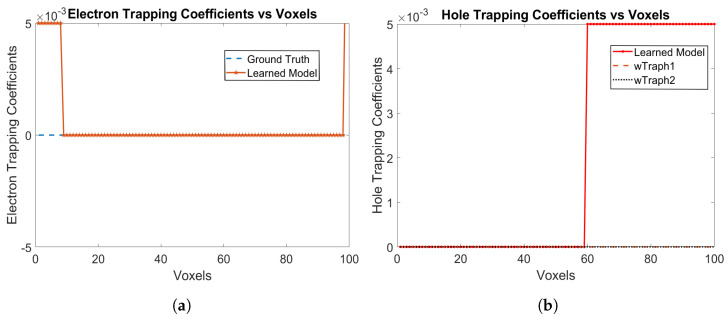
(**a**) Electron trapping coefficients, and (**b**) hole trapping coefficients using Physical Model-3 for e-h injections at Voxels 9, 14, 19, 24, 29, 34, 39, 44, 49, 54 and 59 for the model in Section 3.1.

**Figure 4 sensors-24-00092-f004:**
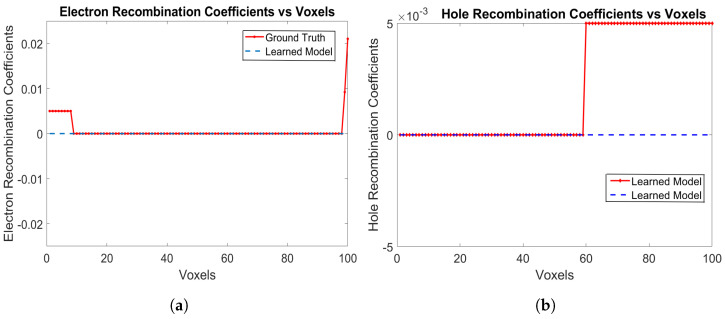
(**a**) Electron recombination coefficients, and (**b**) hole recombination coefficients for e-h injections at Voxels 9, 14, 19, 24, 29, 34, 39, 44, 49, 54 and 59 for the model in Section 3.1.

**Figure 5 sensors-24-00092-f005:**
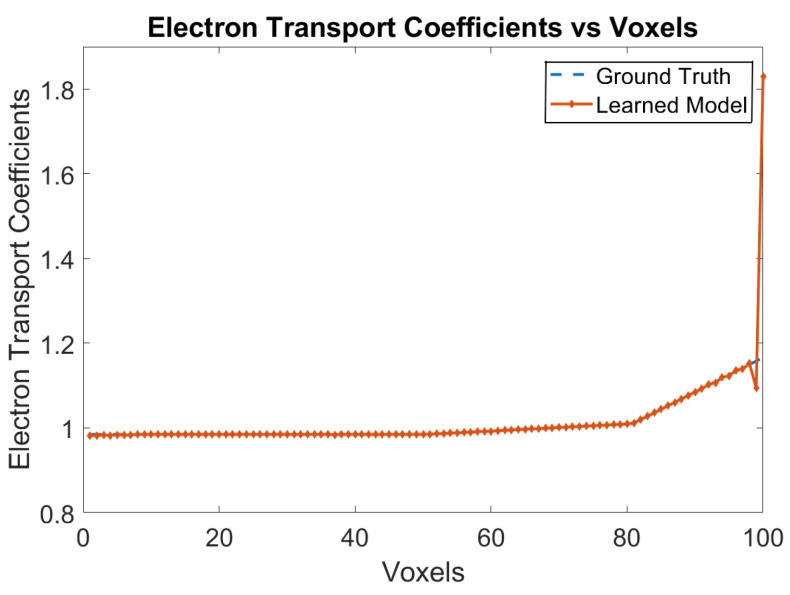
Electron transport coefficients e-h injections at Voxels 9, 14, 19, 24, 29, 34, 39, 44, 49, 54 and 59 for the model in Section 3.2.

**Figure 6 sensors-24-00092-f006:**
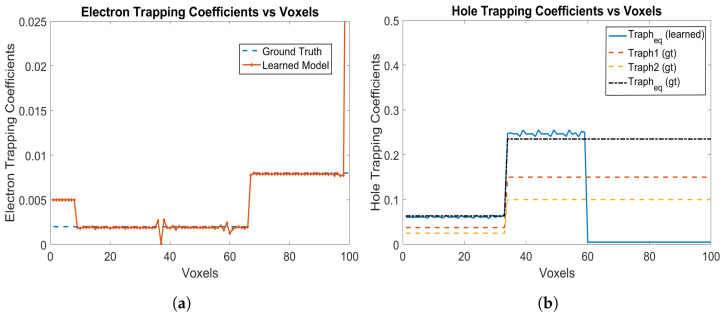
(**a**) Electron trapping coefficients, and (**b**) Hole trapping coefficients using Physical Model-3 for e-h injections at Voxels 9, 14, 19, 24, 29, 34, 39, 44, 49, 54 and 59 for the model in Section 3.2.

**Figure 7 sensors-24-00092-f007:**
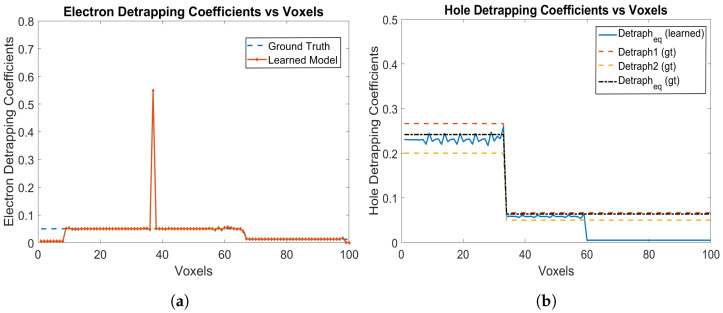
(**a**) Electron detrapping coefficients, and (**b**) hole detrapping coefficients for e-h injections at Voxels 9, 14, 19, 24, 29, 34, 39, 44, 49, 54 and 59 for the model in Section 3.2.

**Figure 8 sensors-24-00092-f008:**
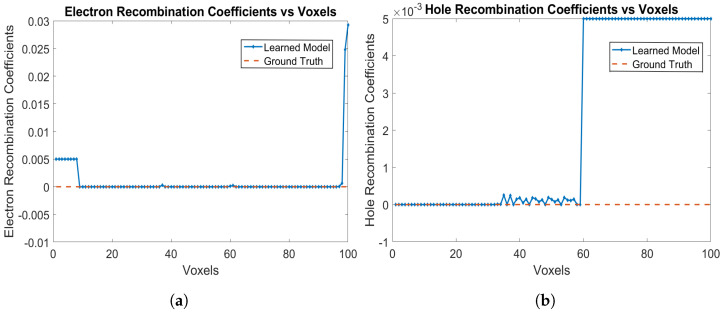
(**a**) Electron recombination coefficients, and (**b**) hole recombination coefficients for e-h injections at Voxels 9, 14, 19, 24, 29, 34, 39, 44, 49, 54 and 59 for the model in Section 3.2.

**Figure 9 sensors-24-00092-f009:**
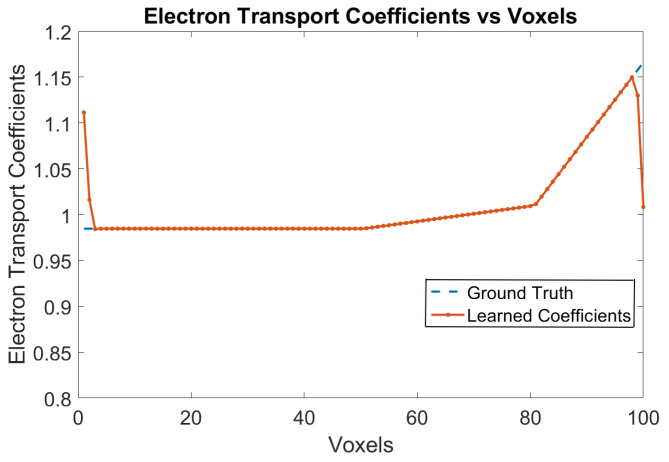
Electron transport coefficients for e-h injections at Voxels 9, 14, 19, 24, 29, 34, 39, 44, 49, 54 and 59 for the model in Section 3.3.

**Figure 10 sensors-24-00092-f010:**
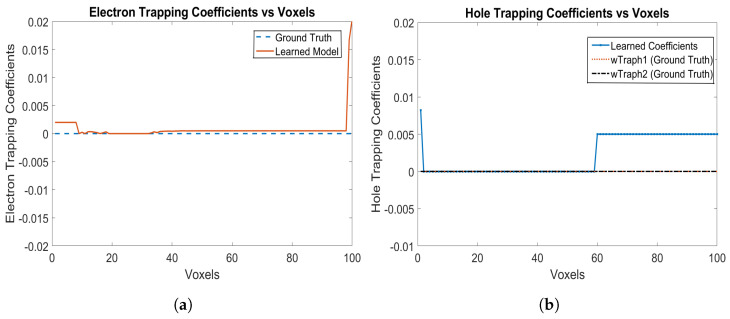
(**a**) Electron trapping coefficients and (**b**) hole trapping coefficients using Physical Model-3 for e-h injections at Voxels 9, 14, 19, 24, 29, 34, 39, 44, 49, 54 and 59 for the model in Section 3.3.

**Figure 11 sensors-24-00092-f011:**
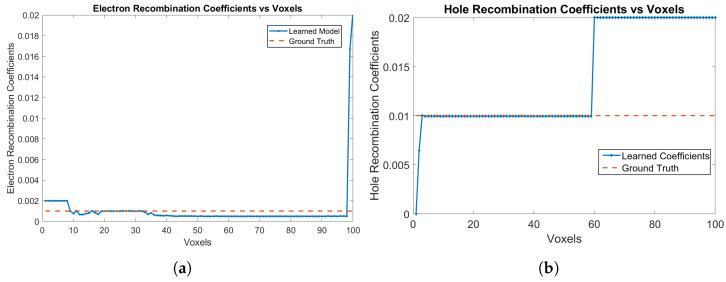
(**a**) Electron recombination coefficients, and (**b**) hole recombination coefficients for e-h injections at Voxels 9, 14, 19, 24, 29, 34, 39, 44, 49, 54 and 59 for the model in Section 3.3.

## Data Availability

All the data and the model in this work will be shared upon reasonable request to the authors. Correspondence and requests for data and model should be addressed to S.B.

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
