# Peer review of "Identifying Defects without a priori Knowledge in a Room-Temperature Semiconductor Detector Using Physics Inspired Machine Learning Model"

_sensors, 2023, doi:10.3390/s24010092_

Round 1

Reviewer 1 Report

Comments and Suggestions for Authors

Authors idea of using Physics Inspired Machine Learning to study the defects centers and recombination is worthful and important contribution in the current RTSD applications towards high energy detectors. However a comparitive analysis using sophesticated analytical techniques with the ML data for electron transport coefficients, electron and hole recombination coefficients and electron hole detrapping coefficients would openup a new insights on the results. However, defect identification using physical models would be of great interest for the reserachers working in the field of RTSD. 

Author Response

We, the authors thank the reviewer for the constructive comments on the submitted manuscript. We address the issues raised by the reviewer -

  1. A comparative analysis using sophisticated analytical techniques with the ML data for electron transport coefficients, electron and hole recombination coefficients and electron, hole detrapping coefficients would open up a new insight on the results.

Answer:-  Experimental approaches in literature such as Thermally Stimulated Conductivity Measurements, Thermoelectric Emission Spectroscopy Measurements, Photon Induced Current Transient Spectroscopy and other methods to characterize the material properties – electron transport coefficients, electron and hole trapping, detrapping and recombination coefficients are done considering uniform material properties in bulk of the material. However, these material properties are non-homogenous at microscopic length scales and our Physics inspired Machine Learning (PI-ML) model characterizes these properties at such small length scales. Experimental approaches in literature are unable to characterize in such small length scales.

On the other hand, using analytical techniques for material characterization in such small length scales is possible by exhaustive search in parameters space. For every combination of material properties (drift, trapping, detrapping and recombination) of electrons and holes, the signals at the electrodes and charges (free and trapped in the voxels) over time can be generated for incident electron-hole injection in every voxel. We can obtain the electrode signals and charges (free and trapped) from experimental setups or from theoretical models. The signals and charges can be correlated using the matched filtering technique, and the best combination of these parameters are identified as the most suitable material properties. This process of exhaustive search combined with matched filtering technique is computationally expensive. For a 100 voxel model, each voxel with 7 parameters, the total number of possible combinations is 107x100 = 10700. Clearly searching over all these values is computationally expensive. Faster algorithms such as Branch and Bound methods can be applied with matched filtering approach. However, searching from a finite set of parameter space is time consuming and would depend on the algorithm used. On the other hand, our PI-ML algorithm is a gradient-based search considering no bounds on the range of material parameters. It is trained in around 4 hours, thereby providing a detailed material characterization in a reasonable time.

Thus, comparative analysis with analytical methods is not useful as per our detailed analysis.

Reviewer 2 Report

Comments and Suggestions for Authors

The authors have display a brand new method for the calculation of the defects in RTSD, which is very interesting. The introduction of machine learning method really pave the way for the further development of the RTSD, which is believed to improve the efficiency of the device design and analysis. However, there are still several misleading points which have been listed below, and are suggested to be tackled before this article can be finally published.

1, What is the exact definition of a voxel, would authors give a more clear clarification, which would be better for readers to understand.

2, Though authors have given clear description of the PI-ML model, however, it is still suggested that the author shall give a more software description about such model, including the basic model such as CNN or other network, training software MATLAB or Python, flow chart to better display the model and algorithms.

3, Ground truth data were acquired with the physical calculation rather than real experiment, why did not get those data from real experiment. The calculation based on physical model can’t be trusted.

4, There seems to be great difference with PI-ML model result and ground truth data, such as in figure 2 and 3, where the authors are suggested to answer two questions about this.

(a)   Which data should be more reliable?

(b)   Why there are such great difference?

5, Compared with physical model calculation, whether the PI-ML model are superior in efficiency and speed, how is its correctness?

6, it would be better if the authors give a exact material as an example to indicate the performance of the PI-ML model.

Comments on the Quality of English Language

acceptable

Author Response

Please see the pdf file with the point-by-point response to the reviewer's comments.

Reviewer 3 Report

Comments and Suggestions for Authors

The authors present a PI-ML model for characterizing defects in TRSD, encompassing trapping, detrapping, and recombination of electrons and holes. Despite the absence of explicitly specified material properties, the learning model demonstrated the capability to identify both present and absent defects and quantify their magnitudes. While a few trained coefficients deviated significantly, most converged accurately, showcasing the effectiveness of the approach. This methodology is applicable to other PI-ML models, providing valuable insights into material defect identification and quantification. To be considered for publication, it is suggested that the authors add sufficient background information about the detector and mechanism. This would make it easier for a general audience to grasp the big picture and the value of this study.

Author Response

We, the authors thank the reviewer for the constructive comments on the submitted manuscript. We address the issues raised by the reviewer -

  1. It is suggested that the authors add sufficient background information about the detector and mechanism. This would make it easier for a general audience to grasp the big picture and the value of this study.

Answer:-  Sufficient background information about the detector and its mechanism has been added in the revised manuscript. This is aimed at providing a big picture to a general audience and add value to the study.

Reviewer 4 Report

Comments and Suggestions for Authors

The article is devoted to describing an approach to searching for defects in radiation detection devices. Defects are understood as a charged center that traps holes or electrons.

The very relevance of such a task raises questions. As a rule, the most relevant tasks are either searching for the electrical and operational characteristics of a device, depending on its design and possible deviations, including defects. Or, on the contrary, searching for the possibility of determining the concentration of defects based on the characteristics of the device.

The model itself is described sparingly in the work. In fact, no physical characteristics of the devices are given. All results are purely model and do not correlate with experiment.

The main disadvantage of such approaches, in my opinion, is the following. If there is some real physical system, the behavior of its model is always different, both due to inaccuracy of the description, and due to computational errors, and due to erroneous assumptions.

For these reasons, corrections introduced into the model (for example, defects) may not have the same effect as predicted theoretically. Continuing to study the model (and not the physical device), the authors will remain captive to these errors. For this reason, the model needs verification and calibration. That is why training samples, or at least test samples, must be determined through experiment.

Thus, there is a certain model in Matlab, the predictions of which can be obtained using machine learning. At the same time, the question remains unanswered whether the locations of defects in a real structure can be calculated. In my opinion, the publication of such an article is not suitable for a journal with high ratings.

Author Response

Please see the pdf file containing point-by-point response, attached to this email.

Round 2

Reviewer 2 Report

Comments and Suggestions for Authors

This time, the authors have addressed the comments, and revised the corresponding sections on the manuscript. So, in my opinion, I suggest its acceptance.

Reviewer 4 Report

Comments and Suggestions for Authors

The authors of the article convincingly answered my questions. In addition, additions made at the request of other reviewers had a positive impact on the overall impression of the article. I still believe that the value and impact of the article are not undoubted, but I have no formal comments on the work.